# Gauge Object Oriented Programming in Student's Learning Performance, Normalized Learning Gains and Perceived Motivation with Serious Games

**Suhni Abbasi** [1,2,*] 🄳, **Hameedullah Kazi** [1] 🄳, **Ahmed Waliullah Kazi** [1], **Kamran Khowaja** [1] and **Ahsanullah Baloch** [1]

1    Department of Computer Science, Faculty of Engineering, Science and Technology, Isra University, Hyderabad 71000, Pakistan; hkazi@isra.edu.pk (H.K.); wali.kazi@isra.edu.pk (A.W.K.); kamran.khowaja@isra.edu.pk (K.K.); ahsanullah.baloch@isra.edu.pk (A.B.)
2    Information Technology Centre, Sindh Agriculture University, Tandojam, Hyderabad 70050, Pakistan
*    Correspondence: suhni.abbasi@sau.edu.pk; Tel.: +92-346-220-6756

**Abstract:** Serious Games (SG) provide a comfortable learning environment and are productive for various disciplines ranging from Science, Technology, Engineering, and Mathematics (STEM) to computer programming. The Object Oriented (OO) paradigm includes objects related to real life, and is considered a natural domain that can be worked with. Nonetheless, mapping those real-life objects with basic Object-Oriented Programming (OOP) concepts becomes a challenge for students to understand. Therefore, this study is concerned with designing and developing an SG prototype to overcome students' difficulties and misconceptions in learning OOP and achieving positive learning outcomes. An experimental evaluation was carried out to show the difference between the experimental group students' performance, who interact with the developed game, and students of the control group, who learn via the traditional instructional method. The experimental evaluations' main finding is that the experimental group's performance is better than the control group. The experimental group's Normalized Learning Gain (NLG) is significantly higher than the control group ($p < 0.005$, paired $t$-test). The evaluation study results show that the developed prototype's perceived motivation on the Instructional Materials Motivation Survey (IMMS) 5-point Likert scale resulted in the highest mean score for attention (3.87) followed by relevance (3.66) subcategories. The results of this study show that the developed SG prototype is an effective tool in education, which improves learning outcomes and it has the potential to motivate students to learn OOP.

**Keywords:** serious games; object-oriented programming; learning outcomes; normalized learning gains; perceived motivation; performance



## 1. Introduction

Computer programming involves algorithm design, code writing, debugging, testing, and implementation and it requires logical reasoning to solve various real-world problems in different situations. For beginners, programming (mainly basic knowledge) is challenging to learn [1], because rote learning is impossible. In the beginning, students cannot solve complex problems; instead, they need to acquire basic concepts to build a higher cognitive understanding of advanced programming concepts. Transitioning from the programming paradigm, such as procedural programming to Object-Oriented Programming (OOP), is also challenging for students [2].

The Object Oriented (OO) paradigm is a more natural domain because OOP's problem domain comprises objects related to real-life objects [3]. However, the mapping of these real-life scenarios with OOP's basic concepts (for example, class, object, attribute, method, message passing, inheritance, polymorphism, and encapsulation) is tough for students to comprehend [4].

Using OOP's approach has many benefits, such as reducing the overall software development time and providing code re-usability and code organization flexibility. However, high-level software development skills are required for learning OOP, and it has become hard for students to understand its underlying concepts clearly in the beginning. Several studies have been conducted to assess the students' difficulties and misunderstandings in OOP learning. In addition, it is imperative to know what barriers make learning programming difficult and how students could learn correctly and efficiently; these issues are not undertaken in developing the learning environment to achieve positive learning outcomes. Therefore, students becoming proficient in computer programming creates the need to identify diverse ways in which the programming problems can be presented and solved. Choosing the correct programming method is important because the boredom and current knowledge in programming concepts may also affect students' learning performance [5]. To provide an excellent learning experience, one should start by conceptualizing OOP basic features and then moving towards the language's technical details. The learning environment, such as learning by playing games, can enhance the learning process [6] and motivate students to learn entertainingly. In addition, games help stimulate the learner's abstract thinking leading to cognitive thinking and further improve their advanced thinking skills [7]. As a result, students would learn happily due to the attractiveness, immersion, and interactive characteristics of computer games. If teachers apply digital games to their curriculum properly, students' performance could be improved [8,9]. Other educational advantages of using computer games for learning include enhancing problem-solving, motivation, retention [3], active student learning, satisfaction [10,11], and advancing learners' ability to adopt new skill levels and support alternative learning styles [4].

Serious Games (SG) refers to a category of games with a clear and specific educational or learning purpose and is not designed and developed primarily for entertainment purposes [12]. SG is more learner-centered, making the learning process more comfortable, fun, and effective, and the whole learning process can be carried out naturally by playing. In games, entertainment and fun are the main attributes that attract people to participate in the learning experience and performance of the students such as positive learning outcomes and Normalized Learning Gain (NLG) could be improved by adequately mapping the course content into the game elements [2,8].

Researchers have made efforts to identify the effect of incorporating SG as an instructional medium and determining how it supports learning outcomes. Moreover, usage of SG methodology can help to develop effective educational games on specific learning topics [10]. As educational games are mainly used to achieve learning outcomes [13], there is little debate about the implications of using games for OOP learning and the various ways of incorporating games into the learning environment.

Earlier studies help provide valuable knowledge regarding learning theory issues, which provides the foundation for developing SG. Smith [9] describes the five categories of learning theories: behaviorism, cognitivism, constructivism, experimentalism, and Socio-contextual theory. These theories are invoked to strengthen the design of educational computer games. The current literature revealed paucity in using learning theories and instructional designs for creating SG prototypes for learning OOP. Due to the high level of complexity involved in OOP subjects, student boredom and dropout rates may increase. For achieving the positive learning outcomes from SG, this study enlightens the development of an SG prototype named OOsg for learning OOP. There are many reasons why OOP was chosen as the primary focus area of this research. Most students encountered severe difficulties in understanding simple OOP concepts. The motivation behind this research was to provide a solution prototype to improve student's performance and motivate them to grasp the OOP concepts in earlier studies.

## 2. Difficulties in Learning OOP

The student's difficulties were identified through a literature review and the investigation on students' performance while learning OOP. The details about these two methods are discussed in the following sections:

### 2.1. Difficulties Identified from Literature Review

The literature review identified the difficulties of learning OOP, faced by the students. The literature comprises tools for identifying the difficulties, OOP concepts (contents covered in the studies), analysis technique, certain OOP problems, and identified issues.

In order to be good programmers, the students skills that the students are required to learn to become good programmers in programming included: algorithm designing, code writing, debugging, testing, and implementation stages. On the other hand, there are certain problems that actually enable students to learn programming that become efficient include mastered the identification of different ways in which of problems presentation and their solutions. Thomasson [14] discussed that the most common difficulty observed was related to Non-Referenced Class fault; it means that students were not able to integrate the classes into the design properly. Other observed challenges were about Non-Existent Classes, attribute identification problem, and issues of cohesion. Or-Bach and Lavy [15] explored the cognitive difficulties and results show that the attributes in the abstract class were only included, however, when the students failed to include methods of any type. Whereas, some students included extra classes (classes that were not related to the solution or could be integrated into the methods and attributes of existing classes). Furthermore, the students missed the necessary class details, placed insignificant attributes within the class, and reduced cohesion.

Sheetz [16] revealed that learning basic OOP concepts, issues of design problems, and programming techniques are difficult for the students. The research results show that learning basic object concepts is the most difficult part for the students other than designing problems and programming techniques. It is also difficult for the students to use or reuse class libraries and distinguishing between the functions of programming language and OOP language. On the basis of analyses, Ragonis [17] shows, the most common problems that surfaced were: difficulty in general picture of program exaction; state changing during execution; the sequence of method invocations related to problem solving. Additionally, method invocation defines the source of parameters' value and the target of the method's return value. The requirement of the input instructions and the connections between the constructor declaration, invocation, and the execution were also listed, that create more difficulty in understating and implementing high-level concepts such as algorithm designing, methods, designing a program, and OOP concepts in contrast with the topics with low-level conceptual difficulty such as understanding the syntax of any language [18]. Even though the current teaching method applies the segmentation to the complex topics into easily understandable pieces, it is still hard for the beginners to leap from understanding to the implementation of concepts [19–21]. Liberman [22] addressed the student's difficulties and misconceptions in the topics related to interfaces, inheritance, and polymorphism. Understanding these topics is undoubtedly difficult but their implementing is also challenging for the students.

### 2.1.1. Difficulties Identified from Students' Investigations

Learning activities are a series of activities designed to enable learners to participate in problem-solving actively. The practical design of these activities ensures that learners are focused while solving them. With reference to a problem scenario, the activities were followed by a set of eight cognitive activities based on OOP concepts that are available here: https://drive.google.com/file/d/1Vcjg9GPeyPn_F-hR01aNu_YOswlcgIaT/view? usp=sharing (accessed on 5 February 2021). The summary of responses obtained from learners are summarized as follows.

### 2.1.2. Difficulties and Misconceptions in Understanding Classes

Most of the students had mistaken to give a reference to some-non-existing classes. The researcher observes that the students did not identify the classes by considering the given problem scenario. Another difficulty that was observed was that students failed to provide the complete identification of the classes. In addition to these difficulties, students also had some misconceptions of conflation between class and attribute or method or object. Some students' solutions included repeated identification of the classes as the same occurrences of classes are written multiple times in the given problem scenario.

### 2.1.3. Difficulties and Misconceptions in Understanding Objects

The majority of the students had shown the wrong instantiates of the classes, in other words, students had difficulty in showing association of an object with a particular class. In addition to these difficulties, students also had some misconceptions about conflation between object and attribute or class. Some student's solutions included repeated identification of the objects, like the same occurrences of objects written multiple times in the given problem scenario.

### 2.1.4. Difficulties and Misconceptions in Understanding Attributes

Students had difficulty in giving the complete identification of attributes or properties for the particular class. Another difficulty observed was that students failed to identify the attributes of all the classes they had previously identified. In addition to these difficulties, students also had some misconceptions of conflation between attributes and class or method or object.

### 2.1.5. Difficulties and Misconceptions in Understanding Methods

Students had difficulty giving the complete identification of methods or behaviors for the particular class. Another difficulty observed was that students failed to identify the methods of all the classes they had previously identified. In addition to these difficulties, students also had some misconceptions of conflation between methods and class or attributes or objects.

### 2.1.6. Difficulties and Misconceptions in Understanding Inheritance

Students had mistakenly provided a reference to some-non-existing hierarchies of the classes. Another difficulty that has been observed is that students failed to provide the complete hierarchies of classes. In addition to these difficulties, students also had some misconceptions about finding the class that may exist in the identified hierarchy, and students sometimes select the same class as both parents and children. Moreover, in many occurrences of the student's solution, it has been observed that they have chosen a child class as a parent class.

For the sake of this research, the difficulties and misconceptions related to understanding the classes, attributes, methods, objects, and inheritance are considered to be overcome, by applying the proposed research method.

## 3. Competencies Required for Mastering OOP

Competence refers to the cognitive abilities and skills that an individual has or may learn to solve a specific problem [23]. Regarding measurement, Klieme [24] stated that competencies are the range of situations or tasks that one needs to be master of, and assessment of those competencies might be done by challenging the student by providing the sample of such (eventually simulated) conditions. The competency model results from such identification that describes and measures the primary competency subjects that an individual must master in a specific topic.

Havenga [25] discusses how high-performance student programmers can facilitate successful computer programs through thinking processes and strategies. The proposed model is intended to assess student skill improvements in initial programming courses.

The score (s) in their model would be attributed to the student's work in the same way that the teachers assign a score for the semester test. This research aims to find the difference between successful programmers and failed programmers, and show that a framework is needed to support novice programmers. However, no evidence is available for the measurement results based on this model as per our knowledge.

Karmer [26] proposed the OOP's competency structure model and evaluation instruments. To measure the student's competencies in OOP, the proposed model includes two major components: (1) a set of candidates for (potentially measurable) competencies, and (2) a category system that supplies a structure for these competencies. The model has four dimensions and sub-dimensions, i.e., 1. OOP knowledge and skills (competencies required for acquiring core programming knowledge and skills); 2. mastering representation (competencies required to understand any system's formal description, such as syntax or the semantics of any programming language); 3. cognitive process (competencies related to the problem-solving stage, such as understanding the problem, determining how to solve the problem, translating the problem into a computer language program, testing, and debugging the problem program); and 4. metacognitive processes (it includes factors like volition, motivation, self-efficacy, perceived understanding, or theoretical values).

The findings obtained from the Ramanauskaite [27] level-based competency structure for OO courses indicate that the distribution of scores and the importance of the proposed e-evaluation process are more similar to the teacher evaluation than the conventional e-assessment system tasks. Instead of displaying the summary score of all skills, you will see the results of many tasks. The author has, nevertheless, implemented the model in any current study frameworks.

Discussing the various competency models, it is observed that none of the models have described how the competencies should be evaluated either in the traditional learning system or by using any game prototype; it is, therefore, researchers who design the competency model themselves. The competency model is also required as input model for developing the research study prototype to evaluate the core competencies of OOP in students. The designed competency model (CM) is presented in Figure 1.

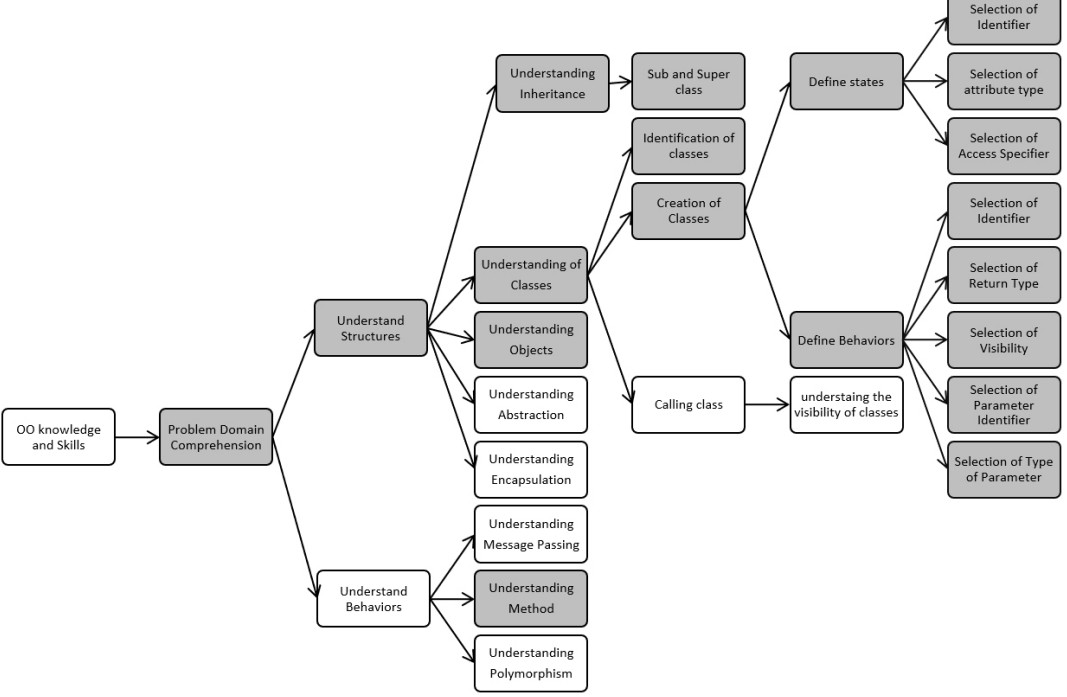

**Figure 1.** Designed competency model for mastering Object-Oriented Programming (OOP) skills.

The designed CM is used to describe the student's competencies that we want to assess, such as knowledge, skills, or other attributes. CM is used primarily to support the reasoning for specific purposes, such as providing scores for students' homework or assignments, certificates, diagnosis, or further guidance. A group of knowledge and skills in CM are called nodes. A more specific CM version is called the student model, which describes competencies at a finer granularity, such as transcripts or progress reports.

This research study emphasizes the achievement of competencies related to the "understand structures" and their underlying sub-competencies. The sub-competencies of "understand structures" which are "understanding inheritance", "understanding of classes", and "understanding of objects" and their sub-competencies (that is, the shaded parts in Figure 1) are focused on in this research study and incorporated in the SG model. The competency "understanding inheritance" has sub-competency or tasks "sub- and super-classes". The node "identification of class" does not include any sub-competency; therefore, it is referred to as concrete tasks/activities. The competency "creation of class" has two sub-nodes related to the "define the status" and "define behaviors"; furthermore, the "define state" and "define behaviors" have other child nodes, considered as concrete task/active nodes.

## 4. Available Tools for Learning OOP

In the past decade, the utilization of SG to learn OOP and other programming-related topics has increased. Many researchers stress the potential of using the SG as initial exposure for learning basic concepts of OOP interactively and engagingly. Mainly, there are two primary purposes for using SG for learning OOP: firstly to promote learning, secondly, motivating, and engaging the students. The example of SG developed for learning OOP can be found in the following sections:

### 4.1. Multiuser Programming Pedagogy for Enhancing Traditional Study (MUPPETS)

Multiuser Programming Pedagogy for Enhancing Traditional Study (MUPPETS) is an immersive OOP learning framework. The framework itself is not a game, but it has an essential mechanism that can provide prompt feedback, interaction, and community integration like modern games. Students learn by creating objects and avatars and sharing the created objects with peers and upper-division students to improve and contribute to students' success. The senior students behave as role models and helpers to the novice. The authors' findings on their student group research praise the strengths of the system. MUPPETS intends to improve the affective and cognitive learning outcomes, students engaged by immediate feedback integrated into the environment, enhancing peer-to-peer collaborations [28].

### 4.2. Alice

Alice [29] is a learner-centric environment that facilitates OOP by allowing objects to be manipulated directly using a limited set of simple commands. In Alice, a simple story can be achieved by selecting objects in the world (such as skaters) and calling one of them (such as Skate). Students can use the graphical interface to incorporate method calls by dragging the name of the method from the object list and placing it in the calling method of the valid location. This textual representation is intended to allow the line of code to be read as a sentence describing the object [30].

### 4.3. Jeliot

Jeliot is an interactive visualization platform designed to help beginners learn procedural and OOP languages [31]. Jeliot supports the method "First Object" or "First Fundamentals" for initial programming courses [32]. Jeliot is primarily intended for OOP Java programming, making it difficult for students to transition between various IDE's to make weaker students feel confused [33].

### 4.4. Greenfoot

Greenfoot is a comprehensive development platform for instructional applications intended to educate young novices. Greenfoot helps to simplify the use of the standard java programming language by creating a personalized environment to minimize much of OOP's difficulty. At the same time, it also adds the ability to create graphics, images, and sounds conveniently so that persuasive examples can be handled as soon as possible. Greenfoot can be used by middle and older students as the first programming method or as the second for young learners [34].

### 4.5. BlueJ

BlueJ is an optimized Java framework for simple OOP concepts [35]. It aims to provide the Java language's easy-to-use teaching environment. However, the source code editor is not technically equivalent to industrial IDEs. Integrating BlueJ with a competent IDE is ineffective. The side effect of using BlueJ in the course too long is that students will use it later to build more complicated projects.

### 4.6. Ztech De

ZTECH is a 2D role-playing game to inspire students to learn OOP in a relaxed, interactive atmosphere. In the game-play, Ztech traveled around the map using the navigation system to battle with enemies to gain experience and win gold points. As the player enjoys the game, they learn OO expertise. The gaming part aims to increase the interest of users in learning the knowledge. The game offers users basic OO concepts such as encapsulation, inheritance, and polymorphism, and other basic programming concepts. Feedback is presented by appropriate dialogue. The authors' findings claimed by testing their game on inexperienced learners include an improvement in learner confidence, courage, and determination to learn and understand OOP concepts [36].

### 4.7. POO SGs

POO SG is a 2D mobile-based game developed for beginners of software development to teach OOP concept basics in a fun and engaging way. The environment of the zoo inspires the gameplay. The game's main purpose is to construct and identify animals and understand various processes, e.g., "voices, acts, attitudes, etc.," by using the OO paradigm. In-game assessment technique is applied to evaluate learners' knowledge about the field in each level of the POO, and the level includes "class, object, inheritance, and polymorphism" OOP concepts. However, the author did not statistically prove the developed game [37].

### 4.8. Discussion on Available Tools for Learning OOP

The OOP was taught through the playing and creation of various SG. The effects of other game-related tools, such as simulators, micro-world, and game engine were also evaluated in addition to the SG. Alice and Jeliot are used as an immersive learning environment in which the code must be used to define the object. Greenfoot, though, is just an educational software creation platform for the use of OOP learning scenarios and it is expected to understand the technical specifics of the code. BlueJ and MUPPETS are optimized learning systems and again need sound computer code skills to learn with these tools. The effect on students observed in terms of results provided by these tools and almost all studies in the study showed results that affect student learning directly or indirectly. Learning from other related instruments, however, includes the order to underscore programming codes that contain all game features like demotivating, boredom, irritation and lack of interest in the topics.

In comparison, ZTECH and POO are SG to learn OOP and have a positive influence on their integration. The studies resulted in cognitive and affected outcomes, learning improvement, problem-solving skills, motivation, student engagement, improved grading, reinforced learning, knowledge acquisition, and student satisfaction effects observed, but still they lack in providing the statistical evidences for the improvement in the learning

outcomes of the students. The studies do not demonstrate the purported link between the game motivation and the actual learning results that are to be obtained with the inclusion of SG. The studies lack in consideration of the difficulties and misconceptions of the student that impede mastering OOP skills. Other problems may include the lack of use of learning theories and instructional design for prototyping SG. Thus, this research acknowledges all of the constraints such as the student's difficulties and misunderstandings, the incorporation of learning theories and instructional design in the design and development of SG prototype. The statistical evidence for the effectiveness of the developed prototype has also been closely analyzed and it provided the relationship between motivation and actual learning outcomes.

## 5. SG Model for Learning OOP

From identifying the student's difficulties and misconceptions through the investigation on student's performance and looking at existing SG available for learning and teaching OOP discussed in Sections 2 and 3, this section stresses the identification of the essential elements that make up an SG. An SG model is required to overcome the difficulties and avoid misconceptions in comprehending given OO problems, such as identifying the classes, their attributes and methods, the impact of creation and destruction of class objects, and establishing the correct hierarchy. The model also aims to foster the learning outcomes and improve the performance of the students.

A model is needed to overcome students' obstacles and follow the entire learning process in a fun and engaging way. Therefore, the development of models is influenced by learning theories, and linking game attributes to instructional design models. For the difficulties arise because of the lack of motivation, the motivational model is incorporated into the model's design. Figure 2 shows an SG model based on the described components, i.e., instructional contents or learning difficulties, game attributes, learning theories, required competencies, and motivational aspects logically placed in the presentation, practice, and performance phases. The purpose of the placement of the components in these phases is presented in Table 1.

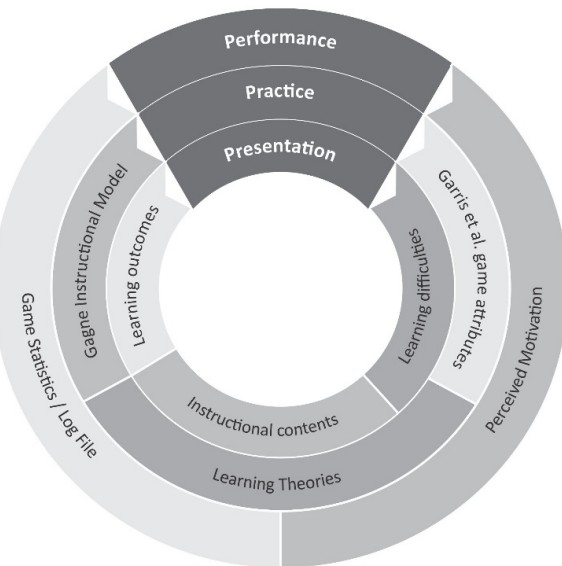

**Figure 2.** Designed Serious Games (SG) model for learning OOP.

**Table 1.** Placement of the components in the SG model's phases.

| Phase | Purpose | Components |
|---|---|---|
| **Presentation** | The input content presented to the user/player as learning material | Instructional contents, learning difficulties and intended learning outcomes |
| **Practice** | Learning and practicing of the user/player | Instructional Models, Learning Theories, Game attributes |
| **Performance** | Assessment of the Performance of the user/player, Game logs/statistics, perceived motivation | |

*5.1. Three Phases of the Model*

The proposed SG model is divided into three phases. The following subsections describe all three phases and their related components.

5.1.1. Presentation

In the presentation phase, the instructional contents of OOP, and the contents on which students faced difficulties and misconceptions collected from existing studies, are presented as a game input or contents to the user/player to learn and improve their performance in those particular OOP tasks.

5.1.2. Practice

In the practice phase, the inputted data from the presentation phase is presented to the user/player in the form of learning activities to practice and accomplish the intended learning outcomes. This stage helps trigger a cycle that includes a sense of achievement or response (such as perceived motivation or interest) or change in user behavior (such as more remarkable persistence or time on task completion), and further system feedback. The practicing phase is the core of the SG model, where the game activities are prudently designed by linking the game attributes with the instructional design model, and the whole learning process has occurred under the rules of the adapted learning theories. Every game activity in the practicing phase is supposed to be played and mastered to achieve one or more learning outcomes and some motivational aspects. The phase components include the instructional design, learning theories, and game attributes.

5.1.3. Performance

Each activity played by the user/player in the practicing phase is assessed to check their performance. This phase includes two types of assessments; first, the formative assessment, where the player is informed about their every correct and incorrect action performed while playing the activities in each game level with the help of log files generated as result of playing game. In the formative assessment, the player is not only informed of their right and wrong actions, but they are also provided with tailored feedback to improve their performance. A summative assessment is done to evaluate the performance of each player, and it is usually done at the end of each level, either completed or not completed by the player. In the summative assessment, the player is informed about their correct and incorrect attempts, remaining actions required to solve the complete level, total time is taken in playing the level, overall score, and their percentage of performance (i.e., correct solution attempted/total correct solution $\times$ 100)

**6. Design and Development of SG for Learning OO: OOsg**

The developed 2D game is named as Object Oriented serious game (OOsg), The motivational point behind the design and development of OOsg is to provide an SG environment, in which the novice programming students or the students who have difficulty in the conceptualization of fundamental quarks of OOP can start learning interestingly and engagingly. The students can improve their concepts, especially the concepts of classes,

attributes, methods, objects, and inheritance using developed SG. The post-test was used to provide the proof of the validity of the learning outcomes using the developed prototype. OOsg starts by providing the personal and control information in which the user has to select the type of player, i.e., basic, intermediate, or expert, to define the size of the solution model. Students may also choose the difficulty level, i.e., simple, medium, or difficult, for defining the time limit for completing the solution, as shown in Figure 3a. The SG incorporated three different stories as playing scenarios, which include the hospital management system, library management system, and the online shopping system. Before starting the game, the students were presented with a warm-up session, as shown in Figure 3b to provide an idea of the environment in which they will play the game. After the warm-up session, the actual game began. Each level of the game begins with a brief introduction to the topic as shown in Figure 3c, and the rules and goals for playing the game as shown in Figure 3d, which were supposed to be achieved at a particular level. The activities provided in the game environment are based on stories related to the problem domain selected at the beginning of the game. Each activity includes a comprehensive learning and assessment program for the student. Every interaction students made with the game is also captured and stored in the game's log file and an increase or decrease in score in the scoreboard, and provided with appropriate and tailored feedback, respectively.

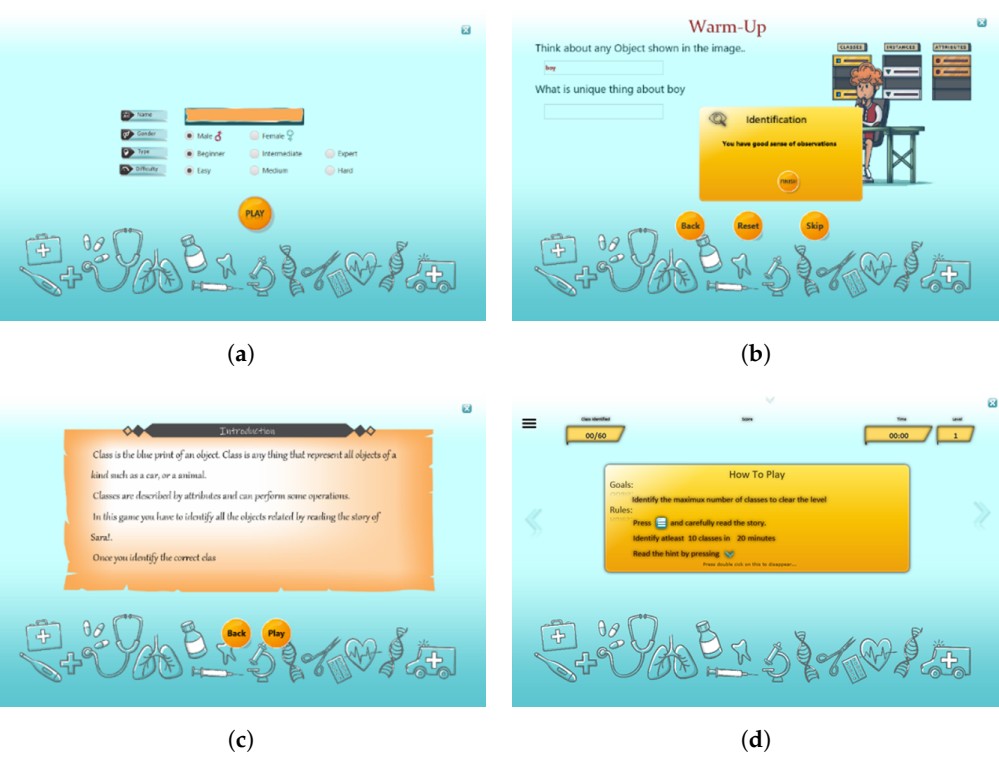

**Figure 3.** Startup of the screen, the screen includes (**a**) personal and game control information, (**b**) warm-up session (**c**), introduction to the basic concepts needed to be learned, and (**d**) rules and goals for playing the levels.

### 6.1. Game Levels

The game includes various levels related to the competencies required to achieve using OOsg. Currently, the game is focused on achieving the shaded competencies shown in Figure 1. Level-1 to level-5 are dedicated for learning about concepts of "class", "attributes", methods, "object", and "relationship between the classes", respectively. In Level-1, shown in Figure 4a, the game story is presented to the player in chunks and the player has to identify the correct candidates for the class according to the domain of the story. The player is supposed to drag the candidate class from the game story and drop it on the empty box available to populate the class list. With the player's correct identification, the

candidate class occurrence would appear in the class list available in the playing region of the screen. In every correct and wrong attempt, the players will be notified with appropriate feedback for their attempt with increment or decrements in the score table. If the player has successfully achieved the game goals, which are set based on the game control information, congratulation messages would greet the players to enhance their motivation to play the game. Once the game control information is matched, the game will go into the stop state. All the actions taken and game statistics are recorded into the log files. Some game level statistics/evaluation information are also presented to the player, such as Win/Lose status, correct and incorrect attempts/solution, how many correct attempts/solutions were remaining, the performance of playing level, total time played, and total score, as shown in Figure 4b. This screen is presented for all the levels if the players play the level until it matches the game control information; otherwise, this information with more details is only recorded into the log files.

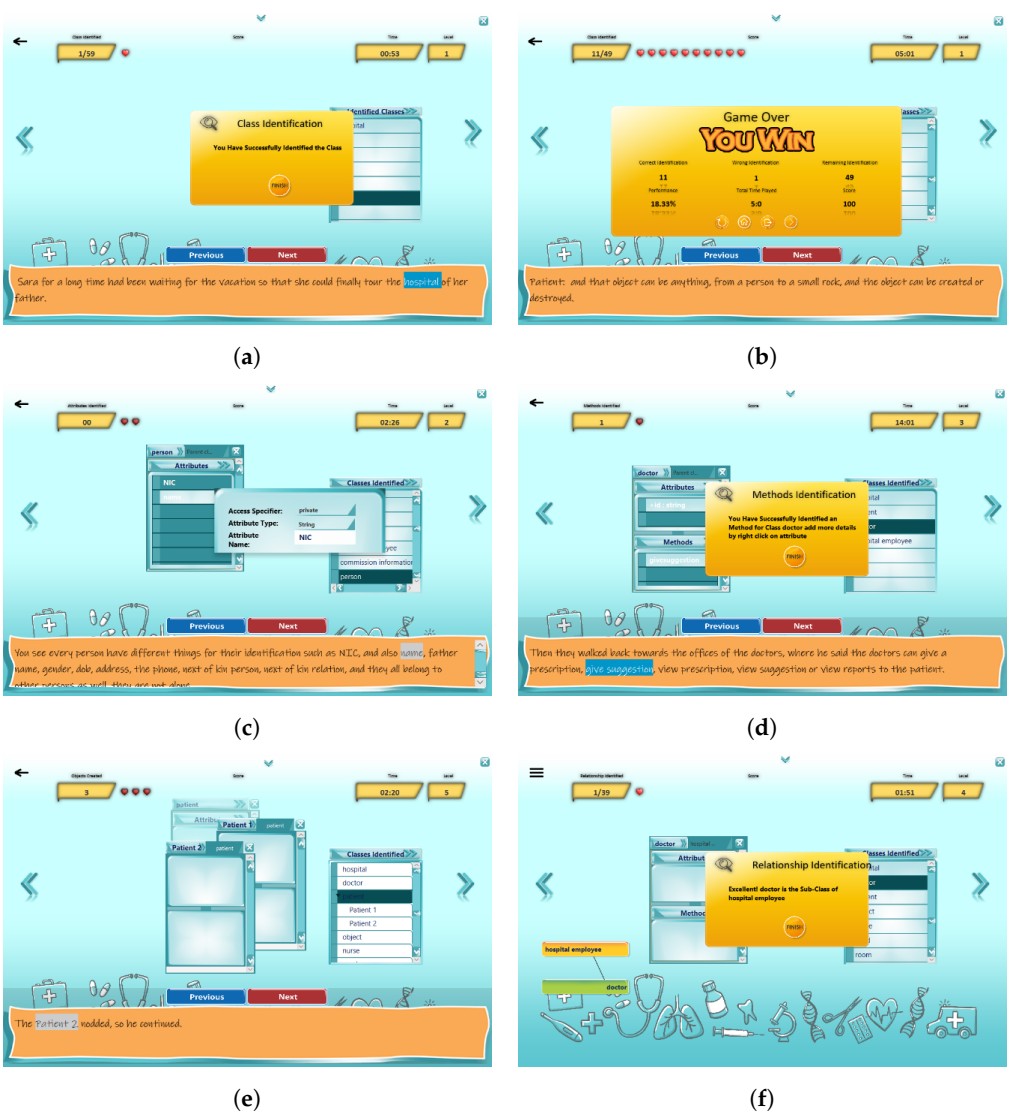

**Figure 4.** Game levels include: (**a**) Level-1—learning about the class identification. Taken from [38]. (**b**) Statistics/evaluation of the Level-1. (**c**) Level-2—learning about the attributes of the identified class. (**d**) Level-3—learning about the methods of the identified class. (**e**) Level-4—learning about the objection creation. (**f**) Level-5—learning about the relationship between classes.

In the Level-2 shown in Figure 4c, the player must identify the attributes for the classes identified in Level-1, whereas, Level-3 shown in Figure 4d is dedicated to the identification of the methods to complete class structure. The class box for each identified class appeared

separately with animation in both levels. If the player clicks on the classes available in the class-list, the box will be highlighted with the class-name on the top of the newly opened class-box, and space for adding the attributes or methods are enabled on each class-box. The increment in the score table and attempt/solution will only be updated once the player completes the details of the identified attributes, i.e., access specifier, attribute type, and attribute name (it is the same as attribute the name identified and written automatically if the player clicks on identified attributes). Initially, the attribute types' combo-box is populated with basic data types and will incrementally be populated with class names, as the class can also be the type of any attribute. However, the strict validation of these details is limited to the scope of this research.

In Level-4, the player is supposed to learn the concept of creation and deletion of the objects and their effects on the class. The player must identify the objects available in the game story or they can create new objects. If the object for a class is available in the game story, and the player drags and drops the object on the relevant class correctly, they will be notified with appropriate feedback as shown in Figure 4e. The player can also create as many objects for a class that they want. However, they have to keep the object name the same as that of the class followed by a digit, e.g., for class "patient", the player can create objects patient1, patient2, and so on. The player will also learn that if an object for a class is destroyed, it will not affect the class itself. However, if any class is destroyed, all its objects will automatically be destroyed.

In the Level-5, the player is supposed to learn the concept of creating the relationship between the classes. The player can activate any class from the class-list and select their parent from the top-right combo-box available on the class-box. The hierarchical relationship between the classes will also appear on the screen's playing region, as shown in Figure 4f. The player can also check the multi-level hierarchy by pressing the class name from the class-list if there is any. The player is also notified with appropriate feedback if they have made any logical mistake for creating the relationship between the classes, such as if the player selects base-class as sub-class or selects the same class both for child and parent-class or if any class is missing between the hierarchy attempt to be made by the player.

### 6.2. Logfile Generated as a Result of playing OOsg

The log files are generated once the user creates their profile shown in Figure 3a. This file keeps all the interactions that players made with the game: the first row shows the variable names for which data are gathered from OOsg, while the remaining rows show the values received for those variables. The example log file for Level-1 is shown in Figure 5; in this example, detailed information about players attempts such as how many time the player attempted to select the attributes, method, or object as a class, what is the total number of classes a player is supposed to identify, and how many classes are remaining will be recorded. The log file records many other concrete details such as total wrong and correct attempts, score, performance (calculated as correct attempts/total No of Solutions × 100), total time played, date and time of playing, and correctly attempted solution. The log file is generated locally for the player of each level in a separate file. The information recorded in the log file helps to evaluate the students, who learned basic OOP concepts using OOsg.

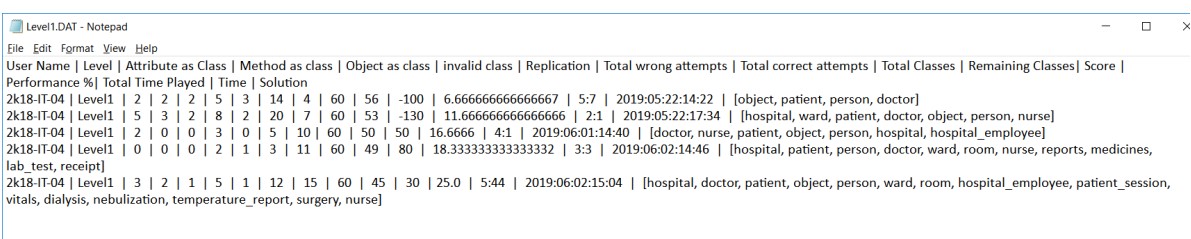

**Figure 5.** Sample log file generated because of interaction with Level-1. Taken from [38].

### 7. Experimental Evaluation and Results

*7.1. Experimental Setup*

The experimental study includes 83 students from local universities of Pakistan, 39 (47%) were male, whereas 44 (53%) were female students. The age reveals 59 (71%) students belonged to the age group of 15–18 years, whereas 24 (29%) students' age ranged from 19 to 24 years, and all were enrolled in various computer science related degree programs. Potential students to participate in the study were selected voluntarily. The students' information about their prior computer programming experience or gaming experience was also asked, because these questions would help to find any potential threats to the validity on pre and post-test scores. The responses show that 61 (74%) participants had no prior programming language experience, whereas 22 (26%) participants had previous programming experience. The responses for prior gaming responses showed that 37 (45%) participants had no prior game experience, whereas 46 (55%) participants had previous gaming experience. Among those 46 (55%) participants, only 12 (14%) participants had prior experience playing educational games, whereas all other participants had experience in other games.

The control and experimental groupings were done according to pre-test scores to ensure that they are grouped according to their OOP performance. After grouping the participants, the intervention session introduces the environment to learn and practice to the control and experimental groups. The control group was treated with the traditional teaching method to grasp the OOP concepts in the classroom setting, whereas, the experimental group was presented with OOsg to interact with. After the intervention sessions, the post-test session started. In the post-test session, the participants were presented with an OOP scenario and questions which needed to be solved in the given time. The scenarios used in pre and post-tests (for information about the pre-and post-test scenarios, and relevant question see Pre and post-test scenarios and associated questions https://drive.google.com/drive/folders/1T5Zk3qSb6cjYhCs79CjNbYwUR014ZG8j?usp=sharing, accessed on 5 February 2021) and the story embedded in the OOsg were designed by keeping the basic quarks of OOP in mind. However, the questions that needed to be solved were kept the same to ensure the responses' accuracy.

In the pre-test learning activity session, all 83 participants were provided with an OOP learning scenario and were asked to answer the given questions. Before the start of the pre-test, the researchers briefly described the purpose of the study. However, students were not informed of the existence of the experimental and control groups. After the pre-test session, the teaching and practice session started with the introduction session in which the domain area of the scenarios needed to be solved, and the procedure to write the solutions was introduced to both groups. After the introduction session, the intervention session begins to learn about the essential feature of OOP. The intervention session is the practice session in which the control group students learn and practice the OOP concepts in the traditional classroom environment. In contrast, the experimental group students were demonstrated with the OOsg to learn and practice by interacting with it. The researcher for both the groups serves as a teacher and facilitator in different sessions. At the end of the intervention session, the post-test session was conducted to assess the learning of students. After the completion of the post-test session, the evaluation session was started. In the evaluation session, the survey form was given to students of both groups separately. The evaluation session was intended to obtain evidence about the effect of using OOsg or traditional teaching methods. There was no time limit set for conducting the evaluation session.

*7.2. Threats to Validity of Findings*

The effectiveness of the OOsg is to obtain sufficient scientific data to provide a detailed explanation of the experimental research findings. Threats to validity are concerns and situations which may misrepresent such facts and thus support (or discard) predicted findings incorrectly. Each threat to validation should be expected and addressed a priori in

order to achieve unbiased outcomes or reduce them with appropriate countermeasures at least. For finding the potential threats, first the Pearson product-correlation coefficient was calculated, but we cannot rely on the R squared value, as we are not checking correlation between two continuous variables. We are checking the correlation between a nominal and continuous variable, so we have calculated the $\eta^2$ by using cross tables. Hence, the following threats to validity were analyzed prior to performing the experimental analysis:

### 7.2.1. Threats of Having Prior Game-Playing Experience

Prior game playing experience may affect the post scores of the students. Hence, Eta values were calculated for a nominal variable "prior game playing experience" and post-test scores of the experimental group. The results of the analysis are provided in Table 2. The $\eta^2$ value is very close to zero, which proves there is no significant effect of student's prior game-playing experience on the experimental group's post-test scores.

### 7.2.2. Threats of Having Prior Computer Programming Experience

Prior computer programming experience may affect the pre and post scores of the students. Hence, Eta values were calculated for a nominal variable "prior computer programming experience", and pre and post-test scores of both the control and experimental groups. The results of the analysis are provided in Table 2. All the $\eta^2$ values for prior computer programming experience and pre and post-test scores of both the control and experimental groups are close to 0, which proves that there is no significant effect of student's prior computer-programming experience on pre and post-test scores of both the groups.

Hence, it is proved that the experimental analysis was unbiased to potential threats to the validity of the results.

**Table 2.** Threats to the validity of the results.

| Var1 | Var2 | r Value | $\eta^2$ |
|---|---|---|---|
| Exp_Group: Post-test score | Prior Game Playing Experience | 0.07 | 0.000 |
| Control_Group: Pre-test score | Prior Programming experience | 0.217 | 0.047 |
| Control_Group: Post-test score | Prior Programming experience | 0.080 | 0.006 |
| Exp_Group: Pre-test score | Prior Programming experience | 0.264 | 0.069 |
| Exp_Group: Post-test score | Prior Programming experience | 0.100 | 0.01 |

### 7.3. Result of Experimental Analysis

The experimental analysis compares the pre and post-test scores of both the experimental and control groups. The details of the studies are discussed in the following section.

### 7.3.1. Measuring Students' Performance for Learning OOP with and without the Intervention of Prototype

In this analysis, both the experimental and control groups' pre-test scores are compared with the post-test scores. The main aim was to determine whether the difference between means for the two sets of scores (pre-test and post-test) is the same or different. The control group revealed that (40) = 0.933, $p > 0.05$, which indicates homogeneous results or no significant difference in the average of the pre-test scores and the post-test scores for the control group. The paired *t*-test results for the experimental group show that t(41) = 14.11, $p < 0.005$, and indicates a significant difference in the average of the pre-test scores and averages of post-test scores. The paired *t*-test result for the control and experimental groups are given in Table 3.

**Table 3.** Measuring student's performance for learning OOP with and without the intervention of the prototype.

| Test-Data | Mean | Statistical Test | t-Value | df | Sig. (2-Tailed) |
|-----------|------|------------------|---------|-----|-----------------|
| Control Group | | | | | |
| Post-Test | 14.341 | Paired *t*-test | 0.933 | 40 | 0.356 |
| Pre-Test | 13.829 | | | | |
| Experimental Group | | | | | |
| Post-Test | 33.285 | Paired *t*-test | 14.11 | 41 | 0.000 |
| Pre-Test | 15.190 | | | | |

### 7.3.2. The Difference in Students' NLG for Learning OOP with and without the Intervention of Prototype

The NLG is the rough measure of the prototype's effectiveness in promoting conceptual understanding of the subject. The amount that students learned was divided by the amount they could have learned [39]. The formula for calculating the normalized gain, proposed by [39], is given below:

$$NLG = (PostTestScore - PreTestScore) \backslash (100 - PreTestScore) \tag{1}$$

For this analysis, each participant's NLG from the control and experimental groups was first calculated by using the Formula (1). The average NLG shown in Figure 6 indicates that the control group shows the NLG was 0.01 or 1% learning gain was found, whereas the experimental groups show 0.21 or 21% gain in the learning of the participants.

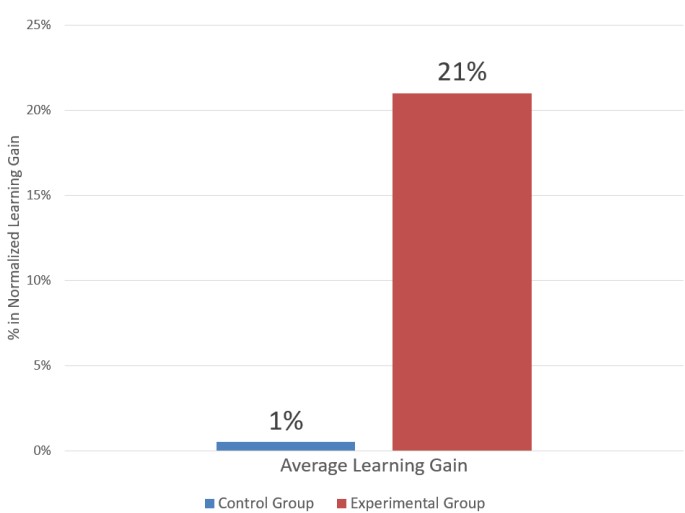

**Figure 6.** Control and experimental group's average Normalized Learning Gain (NLG).

The paired *t*-test results in Table 4 reveal that t(40) = 12.499, *p* < 0.05, which indicates the significant difference between the average NLG of the control and experimental groups.

**Table 4.** The difference in students' NLG for learning OOP with and without the intervention of prototype.

| Test-Data | Mean | Statistical Test | t-Value | df | Sig. (2-Tailed) |
|-----------|------|------------------|---------|-----|-----------------|
| Experimental Group-NLG | 0.2105 | Paired *t*-test | 12.499 | 40 | 0.000 |
| Control Group-NLG | 0.0056 | | | | |

### 7.3.3. The Effect of the Perceived Motivation on Student's Learning

Learner's perception of their motivations refers to how they perceive their motivations while playing OOsg. Although we cannot measure players' direct motivation by any scale, we can estimate players' perceived motivation. Students of both groups evaluated the learner's motivations. The quantitative data were generated using scale Instructional Materials Motivation Survey (IMMS). The four subcategories of the IMMS scale include attention, relevance, confidence, and satisfaction. The mean alpha reliability of their subcategories and overall scale is shown in Figure 7. The quantitative analysis of these subcategories is as follows:

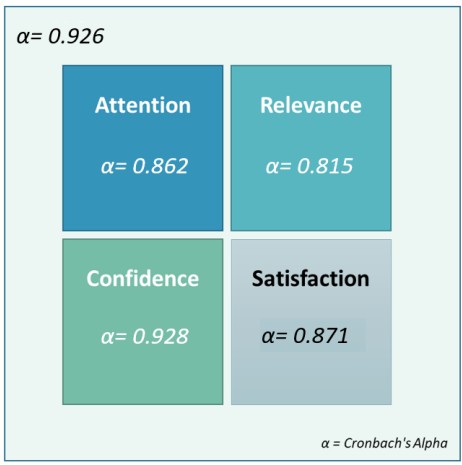

**Figure 7.** Reliability measure for Instructional Materials Motivation Survey (IMMS).

For designing the serious game model, we have first to understand whether a single game attribute leads to learning or enhancing perceived motivation or a combination of multiple attributes within a game has a more substantial effect. In addition, which game elements mainly help to produce which learning outcomes? The details about Gagne's instructional events, OOsg activities, game attributes, and type of the motivational aspects are supposed to be achieved. The attention subcategory is about acquiring and continuously focusing on the learning environment [40], it is a measure of how much students are aware of the instructional design materials used. The game attributes control, and sensory stimuli used in OOsg activities, such as welcoming a player with their chosen name, background music, presenting the game objectives, outcomes, or about information or providing the warm-up scenarios, help in achieving the attention subcategory.

The relevance aims that learning content should not only be accurate, but also consistent with the learning outcomes. The purpose of this category is to determine whether students think that the instructional method, i.e., OOsg or traditional teaching, is related to their existing knowledge, interests, experience, and real life. It shows the learners that their success is a direct result of their efforts and can enhance personal needs and traits related to relevance. They are providing feedback to access learners' efforts which helps in increasing the sense of achievement. In the design of the OOsg, the game attributes, like rules, goals, fantasy, and sensory stimuli used in OOsg activities like providing or presenting the information regarding rules and goals of each level, increase and decrease in points/score, how to get hints, win/lose strategies, help develop relevance between the learning contents and learning outcomes.

Confidence is about students' expectancy of success and learning failure [41]. Keller [42] indicated that confidence is related to the learners' feeling of personal control and its influence on learning effort and performance. In the OOsg, fantasy, rules/goals, sensory stimuli used for presenting the game levels needed to play, performing actions in the levels, or providing feedback on players action taken in the game, helps to develop confidence in the learners.

The last subcategory, satisfaction, is about accomplishments in learning. Several factors can affect satisfaction, such as feedback. Using a comprehensive feedback process, learning iterations, and experiences can support learner self-confidence, maintaining the relationship between attention and learning activities. Establishing clear learning goals can avert adverse effects on learners. Therefore, providing a clear and concise guide will enable learners to make a difference. The game attributes such as fantasy, sensory stimuli, challenge, and mystery provided in activities like providing game score, information for correct and wrong attempts, remaining tasks, etc., or to provide the previous information in the upcoming levels with increased challenge or complexity, help the learners to satisfy their level of satisfaction.

The comparison between the average score for all the IMMS scale subcategories for both the control and experimental groups is shown in Figure 8. The comparison results revealed that the experimental group students' motivation levels were positive for all the subcategories of IMMS.

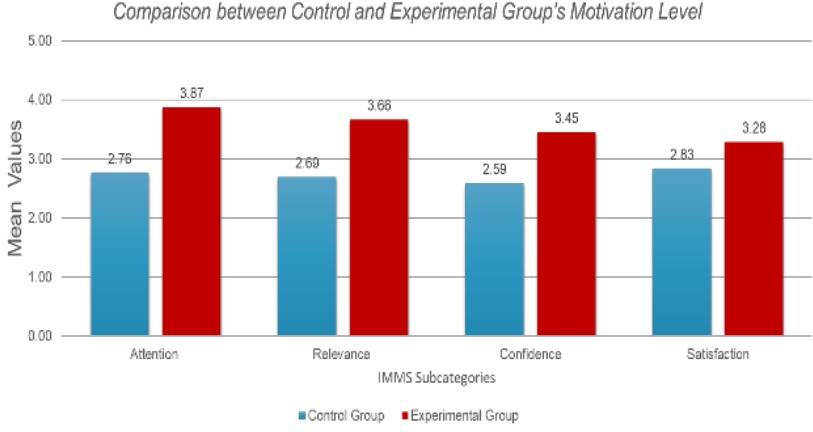

**Figure 8.** Comparison between the control and experimental group's perceived motivational level.

The overall means score, frequency, standard deviation, and percentage of the learner's perception of motivation for the experimental group are presented in Table 5. The highest percentage (47.61%) of the mean score indicates that the majority of the participants have means score between 4.01 to 5.00, for the attention subcategory. The satisfaction subcategory also has the highest percentage (47.61%) for the means score between 3.1 to 4.0. The overall mean score of subcategories of the IMMS scale shows that participants have the high score in attention (3.87) followed by relevance (3.66).

**Table 5.** Overall mean score, frequency, standard deviation, and percentage about the learner's perception of motivation towards learning OOP using OOsg.

| IMMS Subcategories | Mean Score 1.00–2.00 | Mean Score 2.1–3.00 | Mean Score 3.1–4.00 | Mean Score 4.1–5.00 | Mean Score | SD |
|---|---|---|---|---|---|---|
| **Attention** | 1 (2.38%) | 4 (9.52%) | 17 (40.47%) | 20 (47.61%) | 3.87 | 0.65 |
| **Relevance** | 2 (4.76%) | 11 (26.19%) | 13 (30.95%) | 16 (38.10%) | 3.66 | 0.75 |
| **Confidence** | 7 (16.66%) | 6 (14.28%) | 15 (35.71%) | 14 (33.33%) | 3.45 | 0.97 |
| **Satisfaction** | 8 (19.04%) | 5 (11.90%) | 20 (47.61%) | 9 (21.42%) | 3.28 | 1.07 |

## 8. Discussion

This research's primary focus is to design and develop SG prototypes to improve students' performance learning OOP. Initially, students regarded learning programming

languages as a difficult task, which led to low motivation for learning object-oriented and, in many cases, the dropout rate of computer science or related courses was high. Therefore, the prototype was developed and designed to provide an environment where students can learn and practice OOP concepts in a stimulating and engaging way in which the motivation to learn is enhanced without having to worry about failure.

It is discovered that many misconceptions and difficulties hinder the students in learning OOP. The result of reviewed studies showed that many research studies have concluded that programming is difficult for the first time, regardless of the type of programming language [26,43–47] they are learning. In summary, the overall results indicate that students have difficulties and bewilderedness in understanding almost all OOP quarks. However, incorporating all these difficulties into the design and development of SG prototypes is beyond the research scope. Therefore, this study is limited to considering cognitive difficulties related to classes, attributes, behaviors, objects, and hierarchies. Difficulties related to motivation issues are also considered, because the method of teaching OOP concepts was not attractive enough, the complexity of the learning and practice environment was the primary cause for student's lack of interest in learning. Hence, the designed model focused on fostering learning outcomes and improving learning performance by rather starting with the technical details of OOP, students would be provided with an environment where the basic concepts of OOP at the beginning of the course are taught using an entertaining and engaging environment. The students can be motivated to learn boring topics by enjoying the experience. The current literature also revealed paucity in the use of learning theories and instructional designs in the design and development of SG prototypes for learning OOP [3,15,37,48–55]. Despite some promising results obtained from the current literature, it does not show the presumed link between the motivation provided by the games and actual learning outcomes supposed to be achieved by incorporating SG for learning OOP [28].

Therefore, by considering the limitations of the existing SG, the new model was effectively designed. In the formulated model shown in Figure 2, the components discussed in existing SG, i.e., instructional contents or learning difficulties, game attributes, learning theories, required competencies, and motivational aspects, are logically placed in the presentation, practice, and performance phases. The developed SG for learning OOP is named OOsg. To provide the evidence for the effectiveness of OOsg, an experimental evaluation was carried out to analyze the student's performance for learning OO with and without using the developed SG prototype. The pre-test scores of both the experimental and control groups are compared with the post-test scores. The results of the study showed that the experimental group's post-test scores are higher than the pre-test scores. However, no significant difference was found in the post-test and pre-test scores of the control group ($p < 0.005$, paired $t$-test). The average NLG is estimated to measure the effectiveness of the prototype in promoting conceptual understanding of the subject. The result of mean scores for average NLG indicates that only 1% of the gain has been observed for the control group, whereas the experimental groups showed a 21% gain in the participants' learning. The experimental analysis showed that students' NLG was more significant for the experimental group than those of the control group ($p < 0.005$, paired $t$-test).

The evaluation of perceived motivation was conducted using a 5-point IMMS scale. The result showed the highest mean score for attention x = 3.87 followed by relevance x = 3.66 subcategories. The score percentage indicates that the highest percentage of 47.61% for subcategories attention and satisfaction is at a rating of 4.01 to 5.00 and between 3.1 to 4.0, respectively.

## 9. Conclusions

The design and development of the OOsg fulfilled the missing evidence found in existing SG, such as considering the difficulties and misconceptions, incorporation of learning theories, instructional designs, mapping of the game, and learning attributes. The second is the statistical contribution for providing the missing rigorous statistical evidence

between the students who used to learn via traditional teaching methods and the developed prototype. Thus, investigation on the students' difficulties has been done to be considered guidelines for the incorporation as instructional content. As a result, the competency model was designed to determine the major competencies required for mastering the skill of OOP. The competency model is also used as the expected learning outcomes intended to achieve by the developed prototype. Paucity in learning theories, instructional designs, and mapping of the game attributes was expedited by designing a new model for SG. The formulated model is designed based on the extensive review conducted to identify students' difficulties and pitfalls in the existing SG models. OOsg prototype was created to demonstrate the implementation of the designed model. The experimental analysis results showed that the prototype proved to help the students improve the learning outcomes of OOP concepts in which they were facing difficulties.

Through the strong results obtained from the prototypes that have been developed, our work has made an important contribution in encouraging the use of OOsg to learn OO in a fun and engaging environment.

**Author Contributions:** Conceptualization, methodology, validation S.A., H.K., and K.K.; software, visualization, S.A. and K.K.; formal analysis, investigation resources, S.A., H.K., A.W.K., and A.B.; writing—original draft preparation, S.A., H.K., and A.B.; writing—review and editing, H.K., A.W.K., K.K., and A.B.; supervision, H.K., A.W.K., and K.K. All authors have read and agreed to the published version of the manuscript.

**Funding:** This research received no external funding.

**Acknowledgments:** The authors would like to acknowledge postgraduate grant, 17-5/HEC/Sch-Ind/2012 from the Higher Education Commission of Pakistan.

**Conflicts of Interest:** The authors declare no conflict of interest.

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
