# Peer review of "Gauge Object Oriented Programming in Student’s Learning Performance, Normalized Learning Gains and Perceived Motivation with Serious Games"

_information, doi:10.3390/info12030101_

Round 1

Reviewer 1 Report

The authors have carried out an experimental study showing the benefits of learning object-oriented programming by using a game they have developed. The authors have also included a comprehensive review about competencies and other games for learning computer programming. The experimental design and statistical analysis seem to be sound. Finally, conclusions are supported by the results.

However, there are some issues that should be addressed:

- All acronyms must be defined the first time they appear in the text. For instance, OO and OOP in the Abstract.

- References must be numbered in order of appearance in the text. Please note that citations jump from [9] to [32] in section 1.

- Figures do not have enough quality and resolution. Please reconsider to resize Figures 1 and 4. Figures 3 – 7 may also be resized or reorganized. Figure 9 does not read well either.

- Please check the format of the references. In addition, some of them do not include the year of publication.

Reviewer 2 Report

In the paper, the authors present the game teaching elements of the OOP programming and their research concerning the game's usage in the learning process. The idea of the paper is sound.

1) Section 3 presents the existing approaches. There is no summary or comparison of these games. Moreover, I suggest adding some comparison of the proposed game to the existing games.

2) According to the authors: "the formulated model is designed based on the extensive review conducted to identify students’ difficulties and pitfalls in the existing SGs models", however, it is not explicitly listed which difficulties and pitfalls were taken into account. What assumptions were made in creating the game? Please provide the readers with your insight and suggestions for creating such games.

3) In the case of evaluation, please specify how did pre- and post-test look like. How much were these related to the game? Could there be a bias in terms of the gamers and the post-test? 

4) In Section 6.2.3, the authors mention the perceived motivation. Please clarify when (in which moment of the game) and how exactly was it measured.

5) The screenshots presented in Figures: 1, 3d, 5b, 5c, 5d, and 8 are taken from the previous paper of the authors (Abbasi, Suhni, and Hameedullah Kazi. "Stealth assessment in serious games to improve OO learning outcomes." 2019 International Conference on Advances in the Emerging Computing Technologies AECT. IEEE 2020), so these have to be cited to not infringe the publisher's copyright.

6) Although the paper has a comprehensive literature survey, I suggest also adding some most recent directly related sources:
- Molina-Carmona, Rafael, and Faraón Llorens-Largo. "Gamification and advanced technology to enhance motivation in education." Informatics. Vol. 7. No. 2, 2020.
- Flores, Nuno, Ana CR Paiva, and Nuno Cruz. "Teaching Software Engineering Topics Through Pedagogical Game Design Patterns: An Empirical Study." Information 11.3 (2020): 153.
- Peña Miguel, Noemi, Javier Corral Lage, and Ana Mata Galindez. "Assessment of the Development of Professional Skills in University Students: Sustainability and Serious Games." Sustainability 12.3 (2020): 1014.

7) Minor issues:
- There are some typos or other issues:
p. 8 l. 309: indentifiation => indentification
p. 5 l. 179: so you can handle => so one can handle
p. 13 l. 434: can't => cannot
p. 17 l. 532: a SGs => an SGs
- There should be a space before a parenthesis "(" and there should not be a space before a full stop.
- Fig. 8 should be presented as a listing.
- Table 4 is not fully visible.
